# Learning Compact Embedding Layers via Differentiable Product Quantization

## Abstract

Embedding layers are commonly used to map discrete symbols into continuous embedding vectors that reflect their semantic meanings. Despite their effectiveness, the number of parameters in an embedding layer increases linearly with the number of symbols and poses a critical challenge on memory and storage constraints. In this work, we propose a generic and end-to-end learnable compression framework termed differentiable product quantization (DPQ). We present two instantiations of DPQ that leverage different approximation techniques to enable differentiability in end-to-end learning. Our method can readily serve as a drop-in alternative for any existing embedding layer. Empirically, DPQ offers significant compression ratios (14-238x) at negligible or no performance cost on 10 datasets across three different language tasks.

## 1 Introduction

The embedding layer is a basic neural network module which maps a discrete symbol/word into a continuous hidden vector. It is widely used in NLP related applications, including language modeling, machine translation and text classification. With large vocabulary sizes, embedding layers consume large amounts of storage and memory. For example, in the medium-sized LSTM-based model on the PTB dataset (Zaremba et al., 2014), the embedding table accounts for more than 95% of the total number of parameters. Even with sub-words encoding (e.g. Byte-pair encoding), the size of the embedding layer is still very significant. In addition to words/sub-words models in the text domain (Mikolov et al., 2013; Devlin et al., 2018), embedding layers are also used in a wide range of applications such as knowledge graphs (Bordes et al., 2013; Socher et al., 2013) and recommender systems (Koren et al., 2009), where the vocabulary sizes are even larger.

Recent efforts to reduce the size of embedding layers have been made (Chen et al., 2018b; Shu and Nakayama, 2017), where the authors proposed to first learn to encode symbols/words with K-way D-dimensional discrete codes (KD codes, such as 5-1-2-4 for "cat" and 5-1-2-3 for "dog"), and then compose the codes to form the output symbol embedding. However, in Shu and Nakayama (2017), the discrete codes are fixed before training and are therefore non-adaptive and limited to downstream tasks. Chen et al. (2018b) proposes to learn codes in an end-to-end fashion which leads to better task performance. However, their method employs an expensive embedding composition function to turn KD codes into embedding vectors, and requires a distillation procedure which incorporates a pre-trained embedding table as guidance, in order to match the performance of the full embedding baseline.

In this work, we propose a novel differentiable product quantization (DPQ) framework. The proposal is based on the observation that the discrete codes (KD codes) are naturally derived through the process of quantization (product quantization by Jegou et al. (2010) in particular). We also provide two concrete approximation techniques that allow differentiable learning. By making the quantization process differentiable, we are able to learn the KD codes in an end-to-end fashion. Compared to the existing methods (Chen et al., 2018b; Shu and Nakayama, 2017), our framework 1) brings a new and general perspective on how the discrete codes can be obtained in a differentiable manner; 2) allows more flexible model designs (e.g. distance functions and approximation algorithms), and 3) achieves better task performance as well as compression efficiency (by leveraging the sizes of product keys and values) while avoiding the cumbersome distillation procedure.

We conduct experiments on ten different datasets across three tasks, by simply replacing the original embedding layer with DPQ. The results show that DPQ can learn compact discrete embeddings with higher compression ratios than the existing methods, at the same time achieving the same performance as the original full embeddings. Furthermore, our results are obtained from end-to-end training where no extra procedures such as distillation are required. To the best of our knowledge, this is the first work to train compact discrete embeddings in an end-to-end fashion without distillation.

## 2 METHOD

**Problem setup.** An embedding function can be defined as $\mathcal{F}_{\mathcal{W}} : \mathcal{V} \to \mathbb{R}^d$, where $\mathcal{V}$ denotes the vocabulary of discrete symbols, and $\mathcal{W} \in \mathbb{R}^{n \times d}$ is the embedding table with $n = |\mathcal{V}|$. In standard end-to-end training, the embedding function is jointly trained with other neural net parameters to optimize a given objective. The goal of this work is to learn a compact embedding function $\mathcal{F}_{\mathcal{W}'}$ in the same end-to-end fashion, but the number of bits used for the new parameterization $\mathcal{W}'$ is substantially smaller than the original full embedding table $\mathcal{W}$.

**Motivation.** To represent the embedding table in a more compact way, we can first associate each symbol with a K-way D-dimensional discrete code (KD code), and then use an embedding composition function that turns the KD code into a continuous embedding vector (Chen et al., 2018b). However, it is not clear where the discrete KD codes come from. One could directly optimize them as free parameters, but it is both ad-hoc and restrictive. Our key insight in this work is that discrete codes are naturally derived from the process of quantization (product quantization (Jegou et al., 2010) in particular) of a continuous space. It is flexible to specify the quantization process in various ways, and by making this quantization process differentiable, we enable end-to-end learning of discrete codes via optimizing some task-specific objective.

### 2.1 DIFFERENTIABLE PRODUCTION QUANTIZATION FRAMEWORK

The proposed differentiable production quantization (DPQ) function is a mapping between continuous spaces, i.e. $\mathcal{T} : \mathbb{R}^d \to \mathbb{R}^d$. In between the two continuous spaces, there is a discrete space $\{1, \cdots, K\}^D$ which can be seen as discrete bottleneck. To transform from continuous space to discrete space and back, two major functions are used: 1) a *discretization function* $\phi(\cdot) : \mathbb{R}^d \to \{1, \cdots, K\}^D$ that maps a continuous vector into a K-way D-dimensional discrete code (KD code), and 2) a *reverse-discretization function* $\rho(\cdot) : \{1, \cdots, K\}^D \to \mathbb{R}^d$ that maps the KD code into a continuous embedding vector. In other words, the general DPQ mapping is $\mathcal{T}(\cdot) = \rho \circ \phi(\cdot)$.

**Compact embedding layer via DPQ.** In order to obtain a compact embedding layer, we first take a raw embedding and put it through DPQ function. More specifically, the raw embedding matrix can be presented as a Query matrix $\mathbf{Q} \in \mathbb{R}^{n \times d}$ where the number of rows equals to the vocabulary size. The discretization function of DPQ computes discrete codes $\mathbf{C} = \phi(\mathbf{Q})$ where $\mathbf{C} \in \{1, \cdots, K\}^{n \times D}$ is the *KD codebook*. To construct the final embedding table for all symbols, the reverse-discretization function of DPQ is applied, i.e. $\mathbf{H} = \rho(\mathbf{C})$ where $\mathbf{H} \in \mathbb{R}^{n \times d}$ is the final symbol embedding matrix. In order to make it compact for the inference, we will discard the original embedding matrix $\mathbf{Q}$ and only store the codebook $\mathbf{C}$ and small parameters needed in the reverse-discretization function. They are sufficient to (re)construct partial or whole embedding table. In below, we specify the discretization function $\phi(\cdot)$ and reverse-discretization function $\rho(\cdot)$ via product keys and values.

**Product keys for discretization function $\phi(\cdot)$.** Given the query matrix $\mathbf{Q}$, the discretization function computes the KD codebook $\mathbf{C}$. While it is possible to use a complicated transformation, in order to make it efficient, we simply leverage a Key matrix $\mathbf{K} \in \mathbb{R}^{K \times d}$ with $K$ rows where $K$ is the number of choices for each code bit. In the spirit of *product keys* in product quantization, we further split columns of $\mathbf{K}$ and $\mathbf{Q}$ into $D$ groups/subspace, such that $\mathbf{K}^{(j)} \in \mathbb{R}^{K \times d/D}$ and $\mathbf{Q}^{(j)} \in \mathbb{R}^{n \times d/D}$.

We can compute each of $D$ dimensional KD codes separately. The $j$-th dimension of a KD code $\mathbf{C}_i$ for the $i$-th symbol is computed as follows.

$$\mathbf{C}_i^{(j)} = \arg\min_k \text{dist}\left(\mathbf{Q}_i^{(j)}, \mathbf{K}_k^{(j)}\right) \tag{1}$$

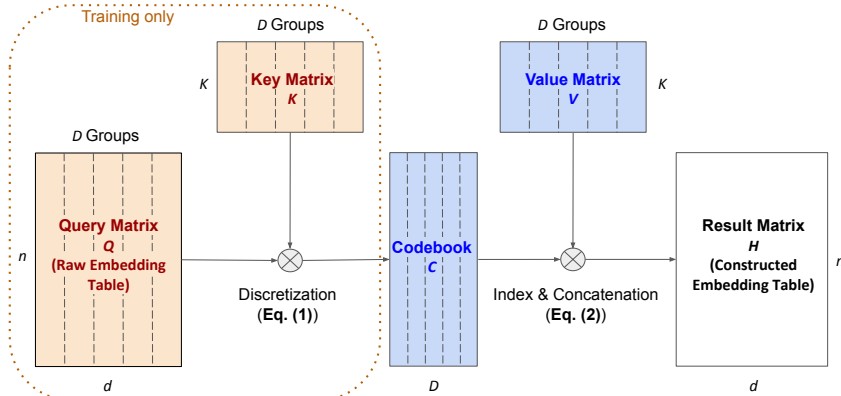

Figure 1: The DPQ embedding framework. During training, differentiable product quantization is used to approximate the raw embedding table (i.e. the Query Matrix). At inference, only the codebook $\mathbf{C} \in \{1, ..., K\}^{n \times D}$ and the Value matrix $\mathbf{V} \in \mathbb{R}^{K \times d}$ are needed to construct the embedding table.

The $\text{dist}(\cdot, \cdot)$ computes distance measure between two vectors, and use it to decide which discrete code to take.

**Product values for reverse-discretization function $\rho(\cdot)$.** Given the codebook $\mathbf{C}$, the reverse-discretization function computes the final continuous embedding vectors. While this can be another sophisticated transformation, we again opt for the most efficient design and employee a single Value matrix $\mathbf{V} \in \mathbb{R}^{K \times d}$ as the parameter. Similarly, we leverage product keys, and split the columns of $\mathbf{V}$ into $D$ groups/subspaces the same way as $\mathbf{K}$ and $\mathbf{Q}$, i.e. $\mathbf{V}^{(j)} \in \mathbb{R}^{K \times d/D}$. We use the code in each of $D$ dimension to index the subspace in $\mathbf{V}$, and concatenate the results to form the final embedding vector as follows.

$$\mathbf{H}_i = [\mathbf{V}^{(1)}_{c_i^{(1)}}, \cdots, \mathbf{V}^{(j)}_{c_i^{(j)}}, \cdots, \mathbf{V}^{(D)}_{c_i^{(D)}}] \tag{2}$$

We note that this is a simplification, both conceptually and computationally, of the ones used in (Chen et al., 2018b; Shu and Nakayama, 2017), which reduces the computation overhead and eases the optimization.

Figure 1 illustrates the proposed framework. The proposed method can also be seen as a learned hash function of finite input into a set of KD codes, and use lookup during the inference instead of re-compute the codes.

**Storage complexity.** Assuming the default 32-bit floating point is used, the original full embedding table requires $32nd$ bits. As for DPQ embedding, we only need to store the codebook and the Value matrix: 1) codebook $\mathbf{C}$ requires $nD \log_2 K$ bits, which is the only thing that depends on vocabulary size $n$, and 2) Value matrix $\mathbf{V}$ requires $32Kd$ bits[1], which does not explicitly depend on $n$ and is ignoble when $n$ is large. Since typically $nD \log_2 K < 32nd$, the DPQ embedding is more compact.

**Inference complexity.** Since only indexing and concatenation (Eq. 2) are used during inference, both the extra computation complexity and memory footprint are usually negligible compared to the regular full embedding (which directly indexes an embedding table).

**Expressiveness.** Although the DPQ embedding is more compact than full embedding, it is not achieved by reducing the rank of the matrix (as in traditional low-rank factorization). Instead, it introduces sparsity into the embedding matrix in two axis: (1) the product keys/values, and (2) top-1 selection in each group/subspace.

**Theorem 1.** *The DPQ embedding matrix $\mathbf{H}$ is full rank given the following constraints are satisfied.*

    1) *One-hot encoded $\mathbf{C} \in \{1, ..., K\}^{n \times D}$, denoted as $\mathbf{B} \in \{0, 1\}^{n \times KD}$, is full-rank.*

    2) *Sub-matrices of splitted $\mathbf{V}$, i.e. $\mathbf{V}^{(j)} \in \mathbb{R}^{K \times d/D}, \forall j$, are all full-rank.*

---

[1] $32Kd/D$ bits if we share the weights among $D$ groups/subspaces.

*3) $KD \geq d$.*

The proof is given in the appendix B. Note that it is easy to keep $\mathbf{H}$ full-rank while achieving good compression ratio, since it is easy to achieve $nD \log_2 K < 32nd$ with $KD = d$.

So far we have not specified some designs of the discretization function such as the distance function in Eq 1. More importantly, *how can we compute gradients through the* $\arg\min$ *function in Eq. 1*? While there could be many instantiations with different design choices, below we introduce two DPQ instantiations that use two different approximation schemes.

## 2.2 SOFTMAX-BASED APPROXIMATION

The first instantiation of DPQ (named DPQ-SX) approximates the non-differentiable $\arg\max$ operation with a differentiable softmax function. To do so, we first specify the distance function in Eq. 1 with a softmax function as follows.

$$\mathbf{C}_i^{(j)} = \arg\max_k \frac{\exp(\langle \mathbf{Q}_i^{(j)}, \mathbf{K}_k^{(j)} \rangle)}{\sum_{k'} \exp(\langle \mathbf{Q}_i^{(j)}, \mathbf{K}_{k'}^{(j)} \rangle)} \tag{3}$$

where $\langle \cdot, \cdot \rangle$ denotes dot product of two vectors (alternatively, other metrics such as Euclidean distance, cosine distance can also be used). To approximate the $\arg\max$, similar to (Chen et al., 2018b; Jang et al., 2016), we relax the softmax function with temperature $\tau$:

$$\tilde{\mathbf{C}}_i^{(j)} = \exp(\langle \mathbf{Q}_i^{(j)}, \mathbf{K}_k^{(j)} \rangle / \tau) / Z \tag{4}$$

where $Z = \sum_{k'} \exp(\langle \mathbf{Q}_i^{(j)}, \mathbf{K}_{k'}^{(j)} \rangle / \tau)$. Note that now $\tilde{\mathbf{C}}_i^{(j)} \in \Delta^K$ is a probabilistic vector (i.e. soft one-hot vector) instead of an integer $\mathbf{C}_i^{(j)}$. And one_hot($\mathbf{C}_i^{(j)}$) $\approx \tilde{\mathbf{C}}_i^{(j)}$, or $\mathbf{C}_i^{(j)} = \arg\max \tilde{\mathbf{C}}_i^{(j)}$. With a one-hot code relaxed into soft one-hot vector, we can replace index operation $\mathbf{V}_{\mathbf{C}_i^{(j)}}^{(j)}$ with dot product to compute the output embedding vector, i.e. $\mathbf{H}_i^{(j)} = \tilde{\mathbf{C}}_i^{(j)} \mathbf{V}^{(j)}$.

The softmax approximated computation defined above is fully differentiable when $\tau \neq 0$. However, to compute discrete codes during the forward pass, we have to set $\tau \to 0$, which turns the softmax function into a spike concentrated on the $\mathbf{C}_i^{(j)}$-th dimension. This is equivalent to the $\arg\max$ operation which does not have gradient.

To enable a pseudo gradient while still be able to output discrete codes, we use a different temperatures during forward and backward pass, i.e. set $\tau \to 0$ in forward pass, and $\tau \to 1$ in the backward pass. So the final DPQ function can be expressed as follows.

$$\mathbf{H}_i = \mathcal{T}(\mathbf{Q}_i | \tau = 1) - \mathrm{sg}\Big( \mathcal{T}(\mathbf{Q}_i | \tau = 1) - \mathcal{T}(\mathbf{Q}_i | \tau = 0) \Big) \tag{5}$$

Where sg is the *stop gradient* operator, which is identity function in forward pass, but drops gradient for variables inside it during the backward pass.

## 2.3 CENTROID-BASED APPROXIMATION

The second instantiation of DPQ (named DPQ-VQ) uses a centroid-based approximation, which directly pass the gradient straight-through (Bengio et al., 2013) a small set of centroids. In order to do so, we need to put $\mathbf{Q}, \mathbf{K}, \mathbf{V}$ into the same space.

First, we treat rows in Key matrix $\mathbf{K}$ as centroids, and use them to approximate Query matrix $\mathbf{Q}$. The approximation is based on the Euclidean distance as follows.

$$\mathbf{C}_i^{(j)} = \arg\min_k \|\mathbf{Q}_i^{(j)} - \mathbf{K}_k^{(j)}\|^2 \tag{6}$$

Secondly, we tie the Key and Value matrices, i.e. $\mathbf{V} = \mathbf{K}$, so that we can pass the gradient through.

We still have the non-differentiable $\arg\min$ operation, and the input query $\mathbf{Q}_i^{(j)}$ are different from selected output centroid $\mathbf{V}_{\mathbf{C}_i^{(j)}}^{(j)}$. However, since they are in the same space, it allows us to directly pass the gradient straight-through as follows.

$$\mathbf{H}_i = \mathbf{Q}_i - \mathrm{sg}(\mathbf{Q}_i - \mathcal{T}(\mathbf{Q}_i)) \tag{7}$$

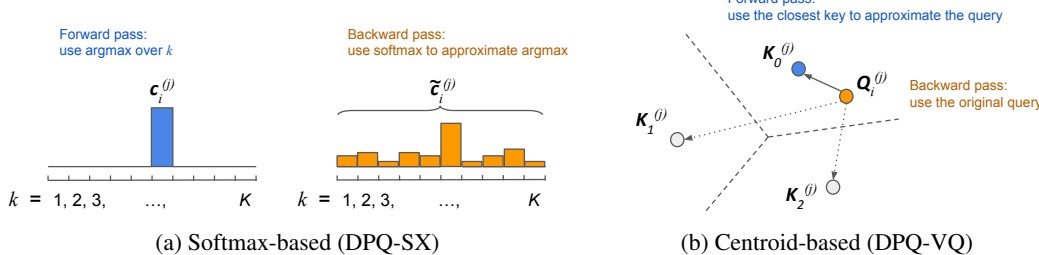

(a) Softmax-based (DPQ-SX)      (b) Centroid-based (DPQ-VQ)

Figure 2: Illustration of two types of approximation to enable differentiability in DPQ.

Table 1: Summary of differences between VQ and SX. DPQ-SX allows more flexibility in distance metrics and whether to tie the Key and Value metrices. DPQ-VQ is more efficient during training and therefore is more scalable to larger $K, D$ values.

| Method | Dist. Metric | Key/Value matrices | Train | Inference |
|--------|--------------|--------------------|-------|-----------|
| DPQ-SX | Dot product and more | Not tied, allows different sizes | Efficient | Efficient |
| DPQ-VQ | Euclidean only | Tied | More efficient | Efficient |

Where sg is again the *stop gradient* operation. During the forward pass, the selected centroid is emitted, but during the backward pass, the gradient is pass to the query directly. This provides a way to compute discrete codes in the forward pass (which are the indexes of the centroids), and update the Query matrix during the backward pass.

However, it is worth noting that the Eq. 7 only approximates gradient for Query matrix, but does not updates the centroids, i.e. the tied Key/Value matrix. Similar to van den Oord et al. (2017), we add a regularization term: $\mathcal{L}_{reg} = \sum_i \|\mathcal{T}(\mathbf{Q}_i) - \text{sg}(\mathbf{Q}_i)\|^2$, which makes entries of the Key/Value matrix arithmetic mean of their members. Alternatively, one can also use Exponential Moving Average (Kaiser et al., 2018) to update the centroids.

**A comparison between DPQ-SX and DPQ-VQ.** DPQ-VQ and DPQ-SX only differ during training. They are very different in how they approximate the gradient for the non-differentiable $\arg\min$ function: DPQ-SX approximates the one-hot vector with softmax, while DPQ-VQ approximates the continuous vector using a set of centroids. Figure 2 illustrates this difference. This suggests that when there is a large gap between one-hot and probabilistic vectors (large $K$), DPQ-SX approximation could be poor; and when there is a large gap between the continuous vector and the selected centroid (large subspace dimension, i.e. small $D$), DPQ-VQ could have a big approximation error.

Table 1 summarizes the comparisons between DPQ-SX and DPQ-VQ. DPQ-SX is more flexible as it does not constrain the distance metric, nor does it tie the Key/Value matrices as in DPQ-VQ. Thus one could use different sizes of Key and Value matrices. Regarding to the computational cost during training, DPQ-SX back-propagates through the whole distribution of $K$ choices, while DPQ-VQ only back-propagates through the nearest centroid, making it more scalable (to large $K, D$, and batch sizes).

## 3 EXPERIMENTS

We conduct experiments on ten datasets across three tasks: language modeling (LM), neural machine translation (NMT) and text classification (TextC) [2] We adopt existing architectures for these tasks as base models and only replace the input embedding layer with DPQ embeddings. The details of datasets and base models are summarized in Table 2.

---

[2]For the five text classification datasets Zhang et al. (2015), Yahoo! answers and AG news represent topic prediction, Yelp Polarity and Yelp Full represent sentiment analysis, and DBpedia represents ontology classification.

Table 2: Datasets and models used in our experiments. More details in Appendix C.

| Task | Dataset | Vocab Size | Tokenization | Base Model |
|---|---|---|---|---|
| LM | PTB
Wikitext-2 | 10,000
33,278 | Words | LSTM-based models from Zaremba et al. (2014), three model sizes |
| NMT | IWSLT15 (En-Vi)
IWSLT15 (Vi-En) | 17,191
7,709 | Words | Seq2seq-based model from Luong et al. (2017) |
| | WMT19 (En-De) | 32,000 | Sub-words | Transformer Base in Vaswani et al. (2017) |
| TextC | AG News
Yahoo! Ans.
DBpedia
Yelp P
Yelp F | 69,322
477,522
612,530
246,739
268,414 | Words | One hidden layer after mean pooling of word vectors, similar to fastText from Joulin et al. (2017) |

Table 3: Comparisons of DPQ variants vs. the full embedding baselines.

| Task | Metric | Dataset | Baseline | DPQ-SX | (CR) | DPQ-VQ | (CR) |
|---|---|---|---|---|---|---|---|
| LM | PPL | PTB
Wikitext-2 | 83.38
95.61 | **83.17**
**94.94** | **(163.2)**
**(59.25)** | 83.27
95.92 | (58.67)
(95.25) |
| NMT | BLEU | IWSLT15 (En-Vi)
IWSLT15 (Vi-En)
WMT19 (En-De) | **25.4**
23.0
**38.8** | 25.3
**23.1**
**38.8** | (86.17)
**(72.00)**
**(18.00)** | 25.3
22.5
38.7 | (16.13)
(14.05)
(18.23) |
| TextC | Acc(%) | AG News
Yahoo! Ans.
DBpedia
Yelp P
Yelp F | **92.59**
69.41
98.12
93.92
**60.33** | 92.49
**69.62**
98.13
**94.17**
60.10 | (19.26)
**(48.16)**
(24.08)
**(38.52)**
(48.16) | 92.55
69.15
**98.14**
93.91
60.22 | (23.95)
(19.24)
**(38.45)**
(24.04)
(24.05) |

We evaluate the models using two metrics: task performance and compression ratio. Task performance metrics are perplexity scores for LM tasks, BLEU scores for NMT tasks, and accuracy in TextC tasks. Compression ratios for the embedding layer is computed as follows:

$$CR = \frac{\text{\# of bits used in the full embedding table}}{\text{\# of bits used in the compressed model during inference}}$$

For DPQ in particular, this can be computed as $CR = \frac{32nd}{nD \log_2 K + 32Kd}$. Further compression can be achieved with 'subspace-sharing' as described in Appendix E.2. With subspace-sharing, $CR = \frac{32nd}{nD \log_2 K + 32Kd/D}$.

### 3.1 COMPRESSION RATIOS AND TASK PERFORMANCE AGAINST BASELINES

Table 3 summarizes the task performance and compression ratios of DPQ-SX and DPQ-VQ against baseline models that use the regular full embeddings[3]. In each task/dataset, we report results from a configuration that gives as good task performance as the baseline (or as good as possible, if it does not match with the baseline) while providing the largest compression ratio. In all tasks, both DPQ-SX and DPQ-VQ can achieve comparable or better task performance while providing a compression ratio from $14\times$ to $163\times$. In 6 out of 10 datasets, DPQ-SX performs strictly better than DPQ-VQ in both metrics. Remarkably, DPQ is able to further compress the already-compact sub-word representations. This shows great potential of DPQ to learn very compact embedding layers.

We also compare DPQ against the following recently proposed embedding compression methods (Chen et al., 2018b; Shu and Nakayama, 2017). **Pre-train**: a three-step procedure where one firstly trains a full model, secondly learns discrete codes to reconstruct the pre-trained embedding

---

[3]For LM, results are from the medium-sized LSTM model.

Table 4: Comparison of DPQ against recently proposed embedding compression techniques on the PTB LM task (LSTMs with three model sizes are studied). Metrics are perplexity (PPL) and compression ratio (CR).

| Method | Small | | Medium | | Large | |
|---|---|---|---|---|---|---|
| | **PPL** | **CR** | **PPL** | **CR** | **PPL** | **CR** |
| Full | 114.5 | 1 | 83.4 | 1 | 78.7 | 1 |
| Pre-train (Chen et al., 2018b) | 108.0 | 4.8 | 84.9 | 11.7 | 80.7 | 18.5 |
| E2E (Chen et al., 2018b) | 108.5 | 4.8 | 89.0 | 11.7 | 86.4 | 18.5 |
| E2E-dist. (Chen et al., 2018b) | 107.8 | 4.8 | 83.1 | 11.7 | **77.7** | 18.5 |
| **DPQ-SX** | **105.8** | **85.5** | **82.0** | **82.9** | 78.5 | **238.3** |
| **DPQ-VQ** | 106.5 | 51.1 | 83.3 | 58.7 | 79.5 | **238.3** |

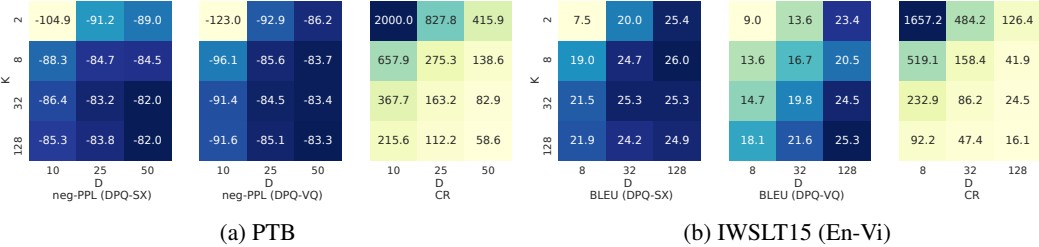

(a) PTB  (b) IWSLT15 (En-Vi)

Figure 3: Heat-maps of task performance and compression ratio for various $K$ and $D$ values. Darker is better. Key observations are: 1) increasing $K$ or $D$ typically improves the task performance at the expense of lower CRs; 2) the combination of a small $K$ and a large $D$ is better than the other way round.

layer and thirdly fixes the discrete codes and trains the model again; **E2E**: end-to-end training without distillation guidance from a pre-trained embedding table; **E2E-dist.**: end-to-end training with a distillation procedure that uses a pre-trained embedding as guidance during training. Table 4 shows the comparison between DPQ and the above methods on the PTB language modeling task using LSTMs with three different model sizes. We find that 1) both Pre-train and E2E achieve good compression ratios but with worse perplexity scores on the Medium and Large models, 2) the E2E-dist. method has the same compression ratio as them and is able to achieve similar perplexity scores as the full embedding baseline, with the downside that it requires the extra distillation procedure, 3) DPQ variants (particularly DPQ-SX) are able to obtain extremely competitive perplexity scores in all cases, while offering compression ratios that are an order of magnitude larger than the alternatives.

### 3.2 EFFECTS OF $K$ AND $D$

Among key hyper-parameters of DPQ are the code size: $K$ the number of centroids per dimension and $D$ the code length. Figure 3 shows the task performance and compression ratios for different $K$ and $D$ values on PTB and IWSLT15 (En-Vi). Firstly, we observe that the combination of a small $K$ and a large $D$ is a better configuration than the other way round. For example, in IWSLT15 (En-Vi), $(K = 2, D = 128)$ is better than $(K = 128, D = 8)$ in both BLEU and CR, with both DPQ-SX and DPQ-VQ. Secondly, increasing $K$ or $D$ would typically improve the task performance at the expense of lower CRs, which means one can adjust $K$ and $D$ to achieve the best task performance and compression ratio trade-off. Thirdly, we note that decreasing $D$ has a much more traumatic effect on DPQ-VQ than on DPQ-SX in terms of task performance. This is because as the dimension of each sub-space $(d/D)$ increases, the nearest neighbour approximation (that DPQ-VQ relies on) becomes less exact.

### 3.3 COMPUTATIONAL COST

DPQ incurs a slightly higher computational cost during training and no extra cost at inference. Figure 4 shows the training speed as well as the (GPU) memory required when using DPQ on the medium

LSTM model, trained on Tesla-V100 GPUs. For most $K$ and $D$ values, the extra training time is within 10%, and the extra training memory is zero. For very large $K$ and $D$ values, DPQ-VQ has better computational efficiency than DPQ-SX (as expected). At inference, we do not observe any impact on speed or memory from DPQ.

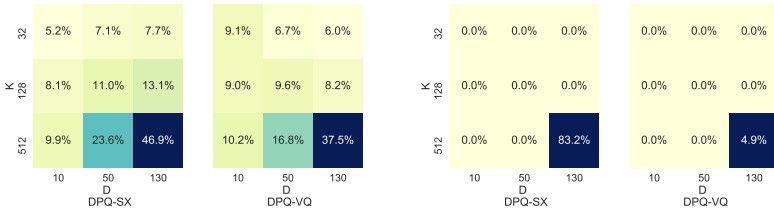

(a) Extra training time used.    (b) Extra training memory used.

Figure 4: Extra training cost incurred by DPQ, measured on a medium sized LSTM for LM trained on Tesla-V100 GPUs. For most $K$ and $D$ values, the extra training time is within 10%, and the extra memory usage is zero. For very large $K$ and $D$ values, DPQ-VQ has better computational efficiency than DPQ-SX in both memory and speed (as expected).

### 3.4 CODE STUDY

To better understand the KD codes learned end-to-end via DPQ, we investigated the codes and observed the following. Firstly, the centroids in all $D$ groups are usually well utilized (Appendix D.1). Secondly, the KD codebook changes as training progresses, but the rate of change decreases throughout training and converges to $< 20\%$ (Appendix D.2). Thirdly, the nearest neighbours in the continuous embedding space between DPQ and the baseline align very well (Appendix D.3). Finally, we also list the learned codes for selected words in Appendix D.4.

## 4 RELATED WORK

Modern neural networks have many parameters and redundancies. The compression of such models has attracted many research efforts (Han et al., 2015; Howard et al., 2017; Chen et al., 2018a). Most of these compression techniques focus on the weights that are shared among many examples, such as convolutional and dense layers (Howard et al., 2017; Chen et al., 2018a). The embedding layers are different in the sense that they are tabular and very sparsely accessed, i.e. the pruning cannot remove rows/symbols in the embedding table, and only a few symbols are accessed in each data sample. This makes the compression challenges different for the embedding layers.

Existing work on compressing embedding layers includes (Shu and Nakayama, 2017; Chen et al., 2018b), which also leverages discrete codes. However, we propose a new formulation from product quantization perspective, in which discrete codes are compute from product quantization on some continuous space. This formulation makes it more general and allows two types of instantiations with different gradient approximation. The product keys and values in our model also make it more efficient in both training and inference. Empirically, DPQ achieve better compression ratios without resorting to the extra distillation process.

Our work differs from traditional quantization techniques (Jegou et al., 2010) in that they can be trained in an end-to-end fashion. The idea of utilizing multiple orthogonal subspaces/groups for quantization is used in product quantization (Jegou et al., 2010; Norouzi and Fleet, 2013) and multi-head attention (Vaswani et al., 2017).

The two approximation techniques presented for DPQ in this work also share similarities with Gumbel-softmax (Jang et al., 2016) and VQ-VAE (van den Oord et al., 2017). However, we do not find using stochastic noises (as in Gumbel-softmax) useful since we aim to get deterministic codes. It is also worth pointing out that these techniques (Jang et al., 2016; van den Oord et al., 2017) by themselves cannot be directly applied to compression.

## 5 CONCLUSION

In this work, we propose a novel and general differentiable product quantization framework for learning compact embedding layers. We provide two instantiations of our framework, which can readily serve as a drop-in replacement for existing embedding layers. Empirically, we evaluate the proposed method on ten datasets across three different language tasks, and show that our method surpasses existing compression methods and can compress the embedding table up to $238\times$ without suffering a performance loss. In the future, we plan to apply the DPQ framework to a wider range of applications and architectures.

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

## A    ALGORITHM PSEUDO-CODE

This section lays out the algorithm pseudo-code for the DPQ embedding layer during the forward training/inference pass.

---

**Algorithm 1** DPQ for the $i$-th token in the vocab (training, forward pass)

---
**h-params** : $K, D$
**parameters** : $\mathbf{Q} \in \mathbb{R}^{n \times D \times (d/D)}$, $\mathbf{K}, \mathbf{V} \in \mathbb{R}^{K \times D \times (d/D)}$, $\mathbf{C} \in \{1, ..., K\}^{n \times D}$

    **for** $j$ in $1, ..., D$ **do**
      $\mathbf{C}_i^{(j)} = \arg\max_k \operatorname{dist}(\mathbf{Q}_i^{(j)}, \mathbf{K}_k^{(j)})$
      $\boldsymbol{h}_i^{(j)} = \mathbf{V}_{\mathbf{C}_i^{(j)}}^{(j)}$
    **end for**
    **return** concatenate($\boldsymbol{h}_i^{(1)}, \boldsymbol{h}_i^{(2)}, ..., \boldsymbol{h}_i^{(D)}$)

---

**Algorithm 2** DPQ for the $i$-th token in the vocab (inference)

---
**h-params** : $K, D$
**parameters** : $\mathbf{V} \in \mathbb{R}^{K \times D \times (d/D)}$, $\mathbf{C} \in \{1, ..., K\}^{n \times D}$

    **for** $j$ in $1, ..., D$ **do**

      $\boldsymbol{h}_i^{(j)} = \mathbf{V}_{\mathbf{C}_i^{(j)}}^{(j)}$

    **end for**
    **return** concatenate($\boldsymbol{h}_i^{(1)}, \boldsymbol{h}_i^{(2)}, ..., \boldsymbol{h}_i^{(D)}$)

---

## B    PROOF OF THEOREM 1

*Proof.* We first re-parameterize both the codebook $\mathbf{C}$ and the Value matrix $\mathbf{V}$ as follows.

The original codebook is $\mathbf{C} \in \{1, \cdots, K\}^{n \times D}$, and we turn each code bit, which is an integer in $\{1, \cdots, K\}$, into a small one-hot vector of length-$K$. This results in the new binary codebook $\mathbf{B} \in \{0, 1\}^{n \times KD}$. Per our constraint in theorem 1, $\mathbf{B}$ is a full rank matrix.

The original Value matrix is $\mathbf{V} \in \mathbb{R}^{K \times d}$, and we turn it into a block-diagonal matrix $\mathbf{U} \in \mathbb{R}^{KD \times d}$ where the $j$-th block-diagonal is set to $\mathbf{V}^{(j)} \in \mathbb{R}^{K \times (d/D)}$. Given that each block diagonal, i.e. $\mathbf{V}^{(j)}$, is full rank, the resulting block diagonal matrix $\mathbf{U}$ is also full rank.

With the above re-parameterization, we can write the output embedding matrix $\mathbf{H} = \mathbf{BU}$. Given both $\mathbf{B}$ and $\mathbf{U}$ are full rank and $KD \geq d$, the resulting embedding matrix $\mathbf{H}$ is also full rank. $\square$

## C    DETAILS OF MODEL TRAINING

We follow the training settings of the base models used, and most of the time, just tune the DPQ hyper-parmeters such as $K$, $D$ and/or subspace-sharing. We also apply batch normalization for the distance measure in DPQ along the K-dimension, i.e. each centroid will have a normalized distance distribution with batch samples.

For training the Transformer Model on WMT'19 En-De dataset, the training set contains approximately 27M parallel sentences. We generated a vocabulary of 32k sub-words from the training data using the SentencePiece tokenizer (Kudo and Richardson, 2018). The architecture is the Transformer Base configuration described in Vaswani et al. (2017) with a context window size of 256 tokens. All models were trained with a batch size of 2048 sentences for 250k steps, and with the SM3 optimizer (Anil et al., 2019) with momentum 0.9 and a quadratic learning rate warm-up schedule with 10k warm-up steps. We searched the learning rate in $\{0.1, 0.3\}$.

## D    CODE STUDY

### D.1    CODE DISTRIBUTION

DPQ discretizes the embedding space into the KD codebook in $\{1, ..., K\}^{n \times D}$. We examine the code distribution by computing the number of times each discrete code in each of the $D$ groups is used in

the entire codebook:

$$\text{Count}_k^{(j)} = \sum_{i=1}^{n} (\mathbf{C}_i^{(j)} == k), \forall j \in \{1, ..., D\}, k \in \{1, ..., K\}$$

Figure 5 shows the code distribution heat-maps for the Transformer model on WMT'19 En-De, with $K = 32$ and $D = 32$ and no subspace-sharing. We find that 1) DPQ-VQ has a more evenly distributed code utilization, 2) DPQ-SX has a more concentrated and sparse code distribution: in each group, only a few discrete codes are used, and some codes are not used in the codebook.

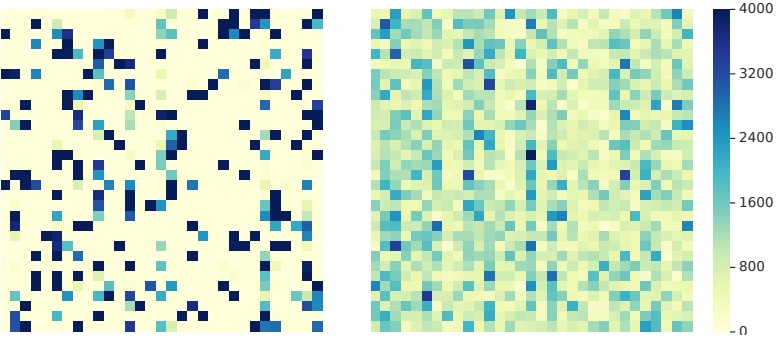

Figure 5: Code heat-maps. Left: DPQ-SX. Right: DPQ-VQ. $x$-axis: K codes per group. $y$-axis: D groups. $K = D = 32$.

## D.2   RATE OF CODE CHANGES

We investigate how the codebook changes during training by computing the percentage of code bits in the KD codebook $\mathbf{C}$ changed since the last saved checkpoint. An example is plotted in Figure 6 for the Transformer on WMT'19 En-De task, with $D = 128$ and various $K$ values. Checkpoints were saved every 600 iterations. Interestingly, for DPQ-SX, code convergence remains about the same for different $K$ values; while for DPQ-VQ, the codes takes longer to stabilize for larger $K$ values.

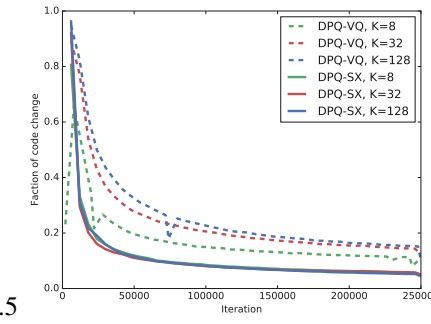

Figure 6: Percentage of code bits in codebook which changed from the previous checkpoint. Transformer on WMT'19 En-De. $D = 128$ for all runs. Checkpoints are saved every 600 iterations.

## D.3   NEAREST NEIGHBOURS OF RECONSTRUCTED EMBEDDINGS

Table 5, 6 and 7 show examples of nearest neighbours in the reconstructed continuous embedding space, trained in the Transformer model on the WMT'19 En-De task. Distance between two sub-words is measured by the cosine similarity of their embedding vectors. Baseline is the original full embeddings model. DPQ variants were trained with $K = D = 128$ with no subspace-sharing.

Taking the sub-word '_evolve' as an example, DPQ variants give very similar top 10 nearest neighbours as the original full embedding: both have 7 out of 10 overlapping top neighbours as the baseline model. However, in DPQ-SX the neighbours have closer distances than the baseline, hence a tighter cluster; while in DPQ-VQ the neighbours are further from the original word. We observe similar patterns in the other two examples.

Table 5: Nearest neighbours of '_evolve' in the embedding space.

| Baseline (Full) | Dist | DPQ-SX | Dist | DPQ-VQ | Dist |
|---|---|---|---|---|---|
| _evolve | 1.000 | _evolve | 1.000 | _evolve | 1.000 |
| _evolved | 0.533 | _evolved | 0.571 | _evolved | 0.506 |
| _evolving | 0.493 | _evolution | 0.499 | _develop | 0.417 |
| _develop | 0.434 | _develop | 0.435 | _evolving | 0.359 |
| _evolution | 0.397 | _evolving | 0.418 | _developed | 0.320 |
| _developed | 0.379 | _arise | 0.405 | _development | 0.307 |
| _developing | 0.316 | _developed | 0.405 | _developing | 0.299 |
| _arise | 0.298 | _resulted | 0.394 | _evolution | 0.282 |
| _unfold | 0.294 | _originate | 0.361 | _changed | 0.278 |
| _emerge | 0.290 | _result | 0.359 | _grew | 0.273 |

Table 6: Nearest neighbours of '_monopoly' in the embedding space.

| Baseline | Dist | DPQ-SX | Dist | DPQ-VQ | Dist |
|---|---|---|---|---|---|
| _monopoly | 1.000 | _monopoly | 1.000 | _monopoly | 1.000 |
| _monopolies | 0.613 | _monopolies | 0.762 | _monopolies | 0.509 |
| monopol | 0.552 | monopol | 0.714 | monopol | 0.483 |
| _Monopol | 0.380 | _Monopol | 0.531 | _Monopol | 0.341 |
| _moratorium | 0.271 | _zugestimmt | 0.486 | _dominant | 0.258 |
| _privileged | 0.269 | legitim | 0.420 | _moratorium | 0.239 |
| _unilateral | 0.262 | _Großunternehmen | 0.401 | _autonomy | 0.230 |
| _miracle | 0.260 | _Eigenkapital | 0.400 | _zugelassen | 0.227 |
| _privilege | 0.254 | _wirkungsvoll | 0.399 | _imperial | 0.226 |
| _dominant | 0.250 | _UCLAF | 0.388 | _capitalist | 0.223 |

Table 7: Nearest neighbours of '_Toronto' in the embedding space.

| Baseline | Dist | DPQ-SX | Dist | DPQ-VQ | Dist |
|---|---|---|---|---|---|
| _Toronto | 1.000 | _Toronto | 1.000 | _Toronto | 1.000 |
| _Vancouver | 0.390 | _Chicago | 0.475 | _Orlando | 0.307 |
| _Tokyo | 0.378 | _Orleans | 0.467 | _Detroit | 0.306 |
| _Ottawa | 0.372 | _Melbourne | 0.435 | _Canada | 0.280 |
| _Philadelphia | 0.353 | _Miami | 0.434 | _London | 0.280 |
| _Orlando | 0.345 | _Vancouver | 0.415 | _Glasgow | 0.276 |
| _Chicago | 0.340 | _Tokyo | 0.407 | _Montreal | 0.272 |
| _Canada | 0.330 | _Ottawa | 0.405 | _Vancouver | 0.271 |
| _Seoul | 0.329 | _Azeroth | 0.403 | _Philadelphia | 0.267 |
| _Boston | 0.325 | _Antonio | 0.400 | _Hamilton | 0.264 |

## D.4 CODE VISUALIZATION

Table 8 shows some examples of compressed codes for both DPQ-SX and DPQ-VQ. Semantically related words share common codes in more dimensions than unrelated words.

## E ADDITIONAL HYPER-PARAMETERS STUDY

### E.1 EFFECTS OF $K$ AND $D$

Figure 7 shows extra heatmaps with varied $K$ and $D$ in addition to those in Section 3.2.

Table 8: Examples of KD codes.

| | DPQ-SX | | | | | | | | DPQ-VQ | | | | | | | |
|---|---|---|---|---|---|---|---|---|---|---|---|---|---|---|---|---|
| _Monday | 2 | 5 | 0 | 7 | 0 | 6 | 1 | 6 | 6 | 5 | 0 | 2 | 4 | 3 | 1 | 7 |
| _Tuesday | 6 | 0 | 0 | 7 | 0 | 6 | 1 | 7 | 1 | 7 | 0 | 2 | 0 | 3 | 1 | 7 |
| _Wednesday | 6 | 5 | 0 | 3 | 0 | 6 | 1 | 6 | 6 | 2 | 3 | 2 | 0 | 2 | 1 | 7 |
| _Thursday | 5 | 5 | 0 | 3 | 0 | 6 | 1 | 7 | 7 | 2 | 0 | 2 | 0 | 3 | 1 | 2 |
| _Friday | 4 | 6 | 0 | 7 | 0 | 6 | 1 | 7 | 6 | 0 | 0 | 2 | 1 | 6 | 1 | 7 |
| _Saturday | 4 | 0 | 6 | 7 | 0 | 6 | 1 | 0 | 6 | 2 | 0 | 2 | 3 | 3 | 1 | 7 |
| _Sunday | 2 | 0 | 0 | 3 | 0 | 6 | 1 | 6 | 7 | 2 | 0 | 2 | 6 | 3 | 1 | 7 |
| _Obama | 2 | 6 | 7 | 2 | 5 | 7 | 3 | 7 | 2 | 3 | 1 | 6 | 6 | 1 | 7 | 4 |
| _Clinton | 2 | 4 | 7 | 2 | 3 | 5 | 6 | 7 | 5 | 3 | 5 | 6 | 6 | 0 | 7 | 4 |
| _Merkel | 4 | 1 | 7 | 2 | 6 | 2 | 2 | 6 | 6 | 3 | 1 | 1 | 4 | 6 | 7 | 4 |
| _Sarkozy | 7 | 6 | 7 | 1 | 4 | 2 | 5 | 0 | 0 | 3 | 1 | 7 | 5 | 7 | 7 | 4 |
| _Berlusconi | 4 | 6 | 5 | 1 | 4 | 2 | 6 | 7 | 6 | 3 | 0 | 6 | 6 | 7 | 7 | 4 |
| _Putin | 2 | 6 | 7 | 1 | 6 | 7 | 6 | 7 | 5 | 3 | 1 | 6 | 6 | 7 | 7 | 6 |
| _Trump | 7 | 6 | 7 | 2 | 0 | 7 | 6 | 7 | 2 | 3 | 1 | 6 | 5 | 7 | 7 | 7 |
| _Toronto | 6 | 2 | 3 | 2 | 4 | 2 | 2 | 6 | 4 | 3 | 4 | 7 | 6 | 2 | 0 | 7 |
| _Vancouver | 2 | 1 | 3 | 2 | 6 | 2 | 5 | 6 | 7 | 3 | 6 | 6 | 6 | 2 | 3 | 1 |
| _Ottawa | 2 | 5 | 6 | 1 | 6 | 2 | 2 | 7 | 6 | 3 | 1 | 6 | 6 | 2 | 0 | 4 |
| _Montreal | 4 | 0 | 0 | 2 | 6 | 2 | 1 | 7 | 4 | 3 | 1 | 1 | 6 | 2 | 0 | 1 |
| _London | 1 | 2 | 0 | 2 | 4 | 7 | 1 | 7 | 2 | 3 | 0 | 2 | 6 | 3 | 3 | 7 |
| _Paris | 4 | 0 | 3 | 5 | 4 | 2 | 1 | 0 | 5 | 3 | 0 | 0 | 6 | 3 | 2 | 7 |
| _Munich | 4 | 2 | 0 | 4 | 0 | 7 | 5 | 0 | 1 | 3 | 3 | 5 | 6 | 3 | 1 | 7 |

(a) WiKiText2       (b) WMT19 (En-DE)

Figure 7: Heat-maps of task performance and compression ratio. Darker is better.

### E.2 SUBSPACE-SHARING

Subspace-sharing refers to the option of whether to share parameters among the $D$ groups in the Key/Value Matrices, i.e. constraining $\mathbf{K}^{(j)} = \mathbf{K}^{(j')}$ and $\mathbf{V}^{(j)} = \mathbf{V}^{(j')}, \forall j, j'$. For simplicity we refer to this as "subspace-sharing". Subspace-sharing improves the compression ratio to: $\text{CR} = 32nd/(nD \log_2 K + 32Kd/D)$.

Figure 8 shows the trade-off curves of task performance and compression ratio with different DPQ variants, K, D and subspace-sharing. We find that one could vary the hyper-parameters to search for optimal performance and compression trade-off. We also observe the effect of subspace-sharing appears very much task-dependent: it improves perplexity scores in LM tasks but hurts BLEU scores in NMT tasks. For TextC tasks, subspace-sharing seems beneficial for DPQ-SX but harmful for DPQ-VQ.

## F RELATIONS TO CHEN ET AL. (2018B) AND OTHER CONVENTIONAL METHODS

Both this work and (Chen et al., 2018b) are based on the idea of representing symbols with discrete codes, but there are some major differences which we listed below:

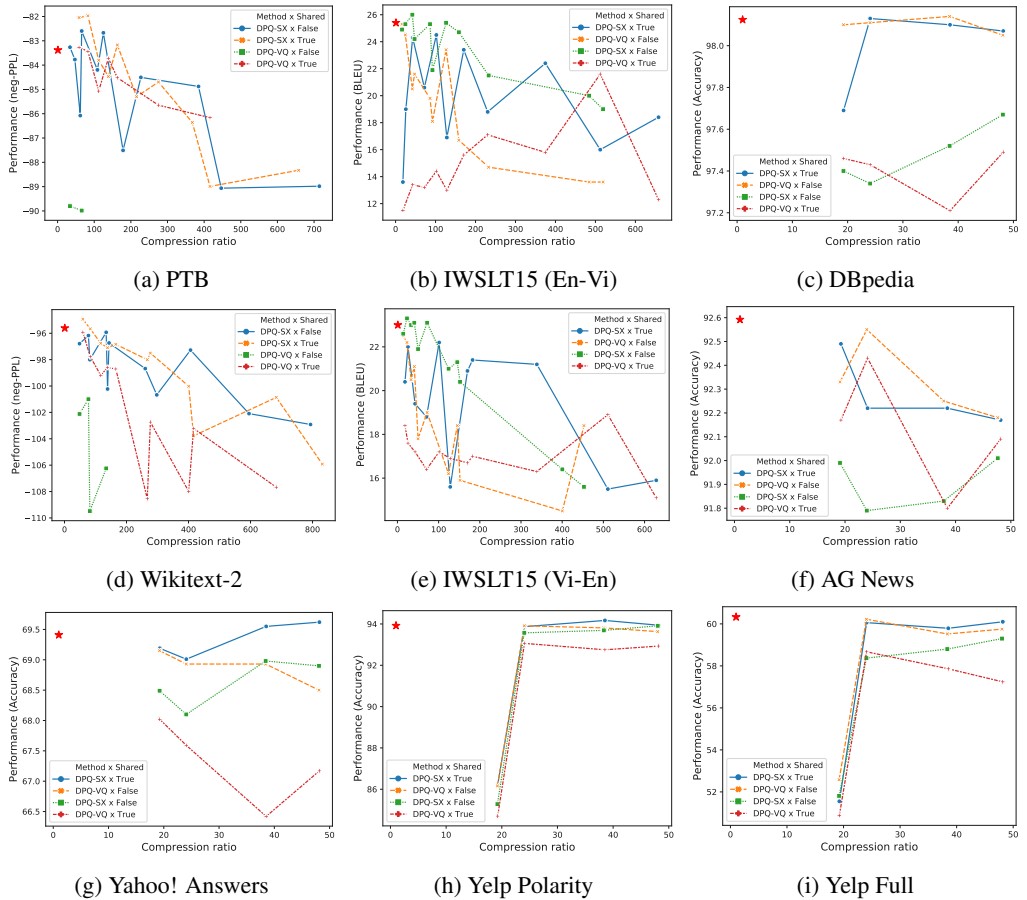

Figure 8: Task performance vs compression ratio trade-off curves. Each subplot comes from one task/dataset and contains four configurations: {DPX-SX, DPX-VQ} × {subspace-sharing, NO-subspace-sharing}.

- In (Chen et al., 2018b), discrete codes are directly associated with each of the symbols, in this work, discrete codes are computed as outcome of product quantization. This shift of perspective allows the proposed framework to generalize beyond a fixed set of vocabulary, and be applied in potentially in any other neural network layers as a stand-alone module.

- Our formulation of discrete codes with product quantization allows us to derive two variants with different approximation techniques (softmax-based and vector quantization-based), while (Chen et al., 2018b) is only based on softmax approximation.

- The product quantization has minimal overhead and is very efficient compared to encoder functions used in (Chen et al., 2018b), i.e. MLP-based and RNN-based functions that compose codes into continuous embedding. Our DPQ has very small memory footprint and computation time overhead (Figure 4). Furthermore, the approximation error are also reduced, and DPQ can be truly trained end-to-end without two pass training with distillation loss as in (Chen et al., 2018b).

Here are comparisons to more traditional approaches:

- Scalar quantization: it quantize each floating number independently, and has very limited compression ratios. E.g. quantizing float32 into int8 would offer a CR of 32/8=4, while likely dropping in task performance metrics (e.g. PPL).

- Product quantization: it generalizes scalar quantization and quantize sub-vectors. However, this approach is non-differentiable (cannot train end-to-end) and requires a post-training procedure. Small quantization errors accumulate and thus performances suffer.

- Pruning: pruning in effect reduces the embedding size for each symbol. Therefore its performance is usually less than ideal (Shu and Nakayama, 2017).
- Low-rank factorization: larger compression ratio requires smaller rank, which in effect reduces embedding table size and leads to worse results.

Different from these techniques, DPQ makes use of discrete codes, and uses product quantization to generate discrete codes. Unlike traditional product quantization, we propose techniques to make it end-to-end differentiable so that the neural nets can adapt to quantization error. DPQ also relates to factorization-based method (Theorem 1), but DPQ can produce high-rank embedding tables with sparse factorization.

## G COMPARISONS TO MORE BASELINES

### G.1 COMPARISONS TO TRADITIONAL COMPRESSION TECHNIQUES

Table 9 shows comparisons on PTB language modeling task (medium-sized LSTM) with broader set of baselines (including methods that are not based on discrete codes). We find that 1) traditional compression techniques, such as scalar and product quantization, as well as low-rank factorization, typically degenerates the performance significantly in order to achieve good compression ratios compared to discrete code learning-based methods (Chen et al., 2018b; Shu and Nakayama, 2017); 2) the proposed method (DPQ) can largely improve the compression ratio while achieving similar or better task performance (perplexity in this case).

Table 9: Performance comparison on PTB language modeling task. The proposed method provides significantly better compression ratio over baselines while achieving similar or better/smaller PPL.

| Method | PPL | Compression ratio |
|---|---|---|
| Full | 83.38 | 1.0 |
| Scalar quantization (8 bits) | 84.06 | 4.0 |
| Scalar quantization (6 bits) | 87.73 | 5.3 |
| Scalar quantization (4 bits) | 92.86 | 8.3 |
| Product quantization(64x325) | 84.03 | 8.3 |
| Product quantization(128x325) | 83.71 | 6.7 |
| Product quantization(256x325) | 83.66 | 5.3 |
| Low-rank (5X) | 84.84 | 5.0 |
| Low-rank (10X) | 85.53 | 10.2 |
| Shu and Nakayama (2017) | 84.92 | 12.5 |
| Chen et al. (2018b) | 83.11 | 12.5 |
| Ours (DPQ-VQ) | 83.3 | 58.7 |
| Ours (DPQ-SX) | **82.0** | **82.9** |

### G.2 COMPARISONS TO BASELINES ON TEXT CLASSIFICATION

Table 10 provides performance comparisons on text classification task. We found that the proposed method (DPQ) usually achieve better accuracies than baselines, at the same time providing better compression ratios.

### G.3 COMPARISONS TO POST-TRAINING RECONSTRUCTION-BASED BASELINES ON NMT

The proposed method (DPQ) supports end-to-end compact embedding learning. An alternative is learning to reconstruct the learned full embedding table with discrete codes after the model is train. The reconstructed compact embedding table is then used to replace the original embedding table for inference. We name this Reconstruction baseline. In our experiment, we use auto-encoder and DPQ (with different $K$ and $D$) to learn to reconstruct the trained full embedding table.

Table 11 shows performance comparisons between the proposed method and reconstruction baseline on WMT19 (En-De) translation task based on Transformer (Vaswani et al., 2017). We can see that

Table 10: Performance comparison on text classification task. The accuracy and compression ratios (in parenthesis) are shown below. The proposed method (DPQ) usually achieve better accuracies than baselines, at the same time providing better compression ratios.

| Dataset | AG News | Yahoo! | DBPedia | Yelp P | Yelp F |
|---|---|---|---|---|---|
| Full | 92.6 (1.0) | 69.4 (1.0) | 98.1 (1.0) | 93.9 (1.0) | 60.3 (1.0) |
| Low-rank(10×) | 91.4 (10.4) | 69.5 (10.2) | 97.7 (10.3) | 92.4 (10.4) | 57.8 (10.3) |
| Low-rank(20×) | 91.5 (21.4) | 69.1 (21.5) | 97.9 (21.3) | 92.4 (21.5) | 57.3 (21.4) |
| Chen et al. (2018b) | 91.6 (53.3) | 69.5 (31.7) | 98.0 (48.4) | 93.1 (48.6) | 59.0 (54.4) |
| DPQ-VQ | **92.6 (24.0)** | 69.2 (19.2) | **98.1 (38.5)** | 93.9 (24.0) | 60.2 (24.1) |
| DPQ-SX | 92.5 (19.3) | **69.6 (48.2)** | 98.1 (24.1) | **94.2 (38.5)** | **60.1 (48.2)** |

the reconstruction baseline degenerates the performance significantly. This is expected as small approximation errors in the embedding layer accumulate and can be amplified as the errors propagate through the deep neural nets, finally lead to large error in output space. Our method does not have this problem as the whole system is jointly trained so the later networks can account for small approximation errors in the early layer.

Table 11: Performance comparisons against the reconstruction baselines.

| Method | BLEU | CR |
|---|---|---|
| Full | 38.8 | 1 |
| Reconstruction (K=128, D=64) | 28.9 | 31.9 |
| Reconstruction (K=32, D=128) | 35.4 | 25.0 |
| Reconstruction (K=128, D=128) | 35.7 | 17.0 |
| Reconstruction (K=32, D=256) | 36.9 | 12.6 |
| Reconstruction (K=128, D=256) | 37.8 | 8.8 |
| DPQ-VQ (K=32, D=128) | **38.7** | **17.0** |
| DPQ-SX (K=32, D=128) | **38.8** | **17.0** |

### G.4 More ablations on DPQ-SX

Table 12 shows an ablation study on PTB language modeling task (medium-sized LSTM), in which we choose to tie the $K$ and $V$ matrices in DPQ-SX (achieved by sharing a single variable during the optimization process). We fix K=128, D=50. We find DPQ-SX (with untied K,V) to perform the best, followed by DPQ-SX (tied K,V) and DPQ-VQ.

Table 12: Ablation study of DPQ-SX on whether or not to tie $K$ and $V$ matrices. By default, DPQ-SX does not tie these two matrices.

| Method | PTB | Wikitext-2 |
|---|---|---|
| DPQ-SX (untied K, V) | **82.4** | **95.2** |
| DPQ-SX (tied K, V) | 83.5 | 95.8 |
| DPQ-VQ | 83.5 | 97.0 |

## H Applying DPQ to BERT

BERT (Devlin et al., 2018) has shown excellent results on a wide range of natural language tasks, therefore it is important to demonstrate that DPQ can also achieve competitive performance on BERT. As our baseline, we pre-trained BERT-base on 512-token sequences for 1M iterations with batch size 1024. We use the same optimizer (Adam) and learning rate schedule as described in Devlin et al. (2018). For the DPQ experiments, we use DPQ-SX with no subspace-sharing, $D = 128$ and $K = 32$; these choices are *not* from hyper-parameter search, but inspired from our results on Transformer on

WMT19 EnDe. We pre-trained BERT with embedding layer replaced by our DPQ, and then finetuned all model parameters for downstream tasks. Note that we do not perform additional tuning for the DPQ experiments in either pre-training or finetuning: we used exactly the same configurations and hyperparameters as in our baseline. Table 13 shows that DPQ performs on par with full embedding in most of the downstream tasks, while giving a compression ratio of $37\times$ on the embedding table. This is equivalent to saving 24M parameters in the BERT-base model, or decreasing the total model size by 22.2%.

Table 13: Effect of using DPQ on BERT. DPQ gives a compression ratio of $37\times$ on the embedding table while the model's performance on downstream tasks remains competitive.

| Embeddings | CR | Squad 1.1 | Squad 2.0 | CoLA | MNLI | MRPC | XNLI |
|---|---|---|---|---|---|---|---|
| Full | 1.0 | 90.1/83.1 | 79.3/76.1 | 81.1 | 84.2 | 86.0 | 53.3 |
| DPQ-SX | 37.0 | 90.0/83.1 | 78.1/74.9 | 80.8 | 83.9 | 85.8 | 53.5 |

