# OpenReview forum: "Learning Compact Embedding Layers via Differentiable Product Quantization"
_ICLR.cc/2020/Conference — Reject_

### Official Review · AnonReviewer3 · 2019-10-11
**Official Blind Review #3**

**Rating:** 3

**Review:**

This paper works on methods for compressed embedding layers for low memory inference, where the compressed embedding are learned together with the task-specific models in a differentiable end-to-end fashion. The methods in the paper build on a slight variant of the K-way D-dimensional discrete code representation proposed by Chen et al.. Specifically, the two methods in the paper are motivated by the idea that the K-way D-dimensional code can be viewed as a form of product quantization. The first proposed method (DPQ-SX) uses softmax-based approximation to allow for differentiable learning, while the second proposed methods uses clustering centroid-based approximation. Empirically, the authors demonstrate that the proposed methods for generating compressed embedding can achieve matching inference accuracy with the corresponding uncompressed full embedding across 3 NLP tasks; these proposed approach can outperform the pre-trained word embedding and the K-way D-dimensional code baselines in language modeling tasks.

This paper builds on an existing embedding representation approach---K-way D-dimensional code. But I think the perspective on viewing k-way D-dim approach as product quantization (which motivate the differentiable learning approach in the paper) is very interesting. Also I think the empirical performance of the proposed method is promising. I gave weak rejection because 1) the proof of theorem 1 is flawed; 2) The experiment might need additional validation to fully support the merit of the proposed methods.

I list below the major concern / questions I have. I am happy to raise the score if the following questions are properly resolved in the rebuttal:

1. Correct me if I am wrong, I think the proof of theorem 1 is wrong---if the integer based code book C is full rank , it does not necessarily imply the one-hot vector based code book B is full rank. E.g. assume K = 2 D = 2,

C = [1 1;1 2;1 1; 1 2] is full-rank (rank = 2), but the corresponding B = [1 0 1 0 ; 1 0  0 1; 1 0 1 0 ; 1 0  0 1] is not full rank (rank < 4).

2. As the proposed methods advocate training the inference-oriented compressed embedding together with the task models (such as translation models), I think the following naive baseline is necessary to fully evaluate the merit of the proposed approach: one can train the full embedding with the task model as usual, compress the task-specific full embedding using the K-way D-dim approach by Chen et al. or using the deep compositional code learning approach by Shu et al., and then use it for the inference. This provides an alternative way to use product quantization based approach for embedding with low inference memory, without training together with task models. Without this, I can not evaluate the necessity to use the train-together approach the author proposed.

3. The proposed DPQ-SX approach performs better in the two proposed approaches. However this approach uses different K and V matrix where in the original K-way D-dim approach we have K = V. This makes it hard to say if the better performance is due to the decoupling of K and V, or because of the training method inspired from the product quantization perspective. It needs ablation study here.

4. In Table 4, the authors only compare to baselines on the LM task, I am wondering how it compares to the the baselines on the other two translation and text classification models.

For improving the paper, the relatively minor comments are as the following:

1. In equation 4, the partition function Z is not explicitly defined.

2. In the second paragraph of section 3.1, it is not clear what exactly is the pre-trained embedding used as baseline.

3. For better readability, it is better to inflate the caption of figure and tables to provide useful take-away message there.


**Experience Assessment:**

I have published one or two papers in this area.

**Review Assessment: Checking Correctness Of Derivations And Theory:**

I carefully checked the derivations and theory.

**Review Assessment: Checking Correctness Of Experiments:**

I carefully checked the experiments.

**Review Assessment: Thoroughness In Paper Reading:**

I read the paper at least twice and used my best judgement in assessing the paper.

---

> ### Author Response · Authors · 2019-11-14
> **Response**
>
> Thank you for your time, careful evaluation and valuable suggestions!
>
> [Theorem 1]
>
> Thank you for pointing this out! We have revised the statement of Theorem 1. Conditioned on B and the sub-matrices of V being full rank, the reconstructed matrix H would be full rank. The original conclusion stays unchanged: despite that DPQ uses fewer bits in its parameterization than the conventional full embedding, its resulting embedding matrix can still be full rank, which is more expressive than factorization methods via low-rank.
>
> [Experiments]
>
> We added more comparisons and ablations in the revision which we hope will address the concerns.
>
> **Post-training embedding compression.**
> As suggested by the reviewer, one could first train the full embedding model and then compress the full embedding table into discrete codes. There are two approaches:
>     1) Fix the discrete codes and re-train the model parameters. This is the "pre-train" method in Paragraph 2 of Section 3.1. Table 4 shows that the performance is not as good as DPQ.
>     2) Use the discrete codes and the reconstructed embeddings directly with the original model (we believe this is what the reviewer suggested). We tested this idea on the NMT task and presented the results in Table 11 in Appendix G. For the same compression ratios, task performance is notably worse than DPQ. This is most likely due to small approximation errors in the embedding layer accumulating over layers.
>
> **Decoupling K and V for DPQ-SX.**
> Figure 12 (Appendix G4) shows that with K=V in DPQ-SX, it incurs a tiny performance (PPL) loss, but still performs slightly better than DPQ-VQ in LM task. Intuitively, we don’t think we should tie K and V for DPQ-SX, as K is used to compute the probability (using dot product as proximity metric) over elements on V. However, it seems the downstream model may be able to adapt to this change of parameterization (and therefore only a slight performance loss).
>
> **More baselines; more tasks.**
> In Table 9 and 10 we show comparison between DPQ and more existing methods on LM and text classification tasks. Our methods outperforms baselines quite significantly and consistently.
>
> **Applying DPQ to BERT.**
> To further verify our method we added experiments on BERT (Appendix H). Without any hyperparameter tuning, DPQ could compress the embedding layer 37x with negligible loss in performance.
>
> [Other]
>
> We also revised the paper to incorporate the minor comments provided by the reviewer. We are happy to address any unresolved concerns or provide more clarifications!

---

> > ### Comment · AnonReviewer3 · 2019-11-15
> > **Thanks for resolving my comments**
> >
> > Dear authors,
> >
> > Thanks for resolving my comments.
> >
> > I think the additional empirical results is more convincing than before to reveal the empirical performance of the new end-to-end embedding compression approach.
> >
> > The only thing that requires some elaboration is the performance of DPQ-SX in PTB in table 9. It is pretty surprising to me DPQ-SX with high compression rate achieves observably better performance than uncompressed embedding. Is it because performance variation due to random seeds? Typically, it requires multiple random seeds to present statistically meaningful results. If the surprising performance is from randomness (such as using only 1 random seed), I would suggest using multiple seeds to enhance the results.
> >
> > In terms of the theorem, I think now the statement makes sense. One typo there is that condition 2) should be V^j \in \mathbb{R}^{K x d / D}.
> >
> > If I understand correctly, the authors aim to say that the newly proposed representation can achieve high rank and preserve more information than low-rank approximation. To enhance the theory, I would suggest the authors to validate that on the embeddings generated by the proposed method, the representation indeed achieves relatively high rank by checking the singular values of the representation. (But as this work is not theory-focused, this suggestion would not dominate my rating.)
> >
> > Given the above, I would raise the rating to boarder line if there is such an option. The limiting factor on rating is the incremental novelty over the existing works such as Chen et al. 2018.

---

> > > ### Author Response · Authors · 2019-11-15
> > > **Thanks for the prompt response!**
> > >
> > > Dear reviewer,
> > >
> > > Thanks for the prompt response!
> > >
> > > We think it is reasonable that our method can achieve better task performance than full embedding in certain tasks/datasets, because DPQ can implicitly regularize the model with more efficient parameterization. We have observed this phenomenon consistently on some tasks (LM on PTB, Wikitext-2) with multiple runs and different random seeds (excluding the possibility of noises).
> > >
> > > Thanks again for another nice suggestion. We manually checked the rank of the reconstructed embedding matrix with numerical method on the NMT/WMT-19(EN-DE) and it is indeed full rank (512 for 32000x512 matrix).
> > >
> > > We firmly believe our work is novel and can make a positive contribution to the community. The major novelty is to **formulate discrete codes via Product Quantization and making it generally end-to-end differentiable**.
> > >
> > > It only bears resemblance to (Chen et al, 2018b) in terms of using discrete codes to tackle the embedding compression problem. However our formulation and training techniques are very different. The product quantization formulation is far more flexible and efficient than (Chen et al, 2018b): it allows optimization with various approximation techniques (SX, VQ), and also allows one-pass end-to-end training whereas (Chen et al, 2018b) is still constrained by an extra distillation procedure. This new formulation led to SOTA empirical results by a large margin. Potentially it can also be applied beyond embedding layer compression (e.g. for dense or conv layers via end-to-end product quantization of weights).
> > >
> > > Please feel free to let us know if there are any remaining concerns, we are happy to further clarify!

---

### Official Review · AnonReviewer1 · 2019-10-16
**Official Blind Review #1**

**Rating:** 6

**Review:**

In this manuscript, authors improve the work in [1] by simplifying the reverse-discretization function. The empirical study demonstrates the effectiveness of the proposed algorithm.
I’m not familiar with the area. The differences between this work and [1] should be elaborated more in the related work, since they are closely related.
Besides, for the technical part, DPQ-SX outperforms DPQ-VQ while the softmax approximation seems identical to that developed in [1].

[1] ICML’18: Learning K-way D-dimensional Discrete Codes for Compact Embedding Representations

**Experience Assessment:**

I do not know much about this area.

**Review Assessment: Checking Correctness Of Derivations And Theory:**

I assessed the sensibility of the derivations and theory.

**Review Assessment: Checking Correctness Of Experiments:**

I assessed the sensibility of the experiments.

**Review Assessment: Thoroughness In Paper Reading:**

I read the paper at least twice and used my best judgement in assessing the paper.

---

> ### Author Response · Authors · 2019-11-14
> **Response**
>
> Thanks for your time and the constructive feedback! While both this work and [1] are based on the idea of representing symbols with discrete codes, the two present very different methodologies. Here are the major differences:
>
>   1) In their work, discrete codes are directly associated with each of the symbols; in this work, discrete codes are computed as outcome of product quantization.
>   2) Our formulation of discrete codes with product quantization allows us to derive two variants with different approximation techniques (softmax-based and vector quantization-based).
>   3) Our method uses a novel composition function (inspired by product quantization), which is much more efficient than before (smaller memory footprint and less computation time overhead, Figure 4).
>   4) With these improvements, DPQ can be trained in a truly end-to-end fashion to achieve an order of magnitude higher compression ratios at negligible or no performance cost.
>
> We have also elaborated these differences in our related work section. Thank you for this suggestion.
>
> Regarding the comparisons between DPQ-SX and DPQ-VQ, they represent two ways of approximating discrete code learning. Each has its advantages and drawbacks (Table 1). In our experiments, we found DPQ-SX performance marginable better than DPQ-VQ for more tasks/datasets, while DPQ-VQ is more computationally efficient during training. There are potential ways to improve compression results for DPQ-VQ in the future, so we believe both variants have their merits.

---

### Official Review · AnonReviewer2 · 2019-10-27
**Official Blind Review #2**

**Rating:** 3

**Review:**

This paper considers the problem of having compact yet expressive KD code for NLP tasks. The authors claim that the proposed differentiable product quantization framework has better compression but similar performance compared to existing KD codes.The authors present two instances of the DPQ framework: DPQ-SX using softmax to make it differentiable, and DPQ-VQ using centroid based approximation. While DPQ-SX performs better in terms of performance and compression, DPQ-VQ has the advantage in scalability.

- Significance
It's understandable that the size of the embedding is important, but there's been a lack of explanation as to why this should be done only through KD codes. Hence, it is doubtful how big the impact of the proposed framework is.

- Novelty
Just extending and making Chen et al., 2018b's distilling method to be differentiable has limited novelty.

- Clarity
The paper is clearly written in most places, but there were some questions about the importance and logic of statements.

- Pros and cons
Compared to Chen et al., 2018b, there is no need to use expensive functions, and performance is better. But, the baseline consists only of algorithms using KD codes; there might be many disadvantages compared to other types of algorithms.

- Detailed comments and questions
1. It is true that the parameters for embedding make up a large part of the overall parameters, but I would like some additional explanation of how important they are to learning. It is usually not necessary to train the entire embedding vector on GPU, so it would not be a big issue in the actual learning process.
2. In a similar vein, it would be nice to show which of the embedding vector size or the LSTM model size contributes significantly to performance improvements. If LSTM model size contributes more, the motivation would be weakened.
3. It would be nice to add more baselines such as Nakayama 2017 as well as the standard compression/quantization methods used in other deep networks. And please explain why we should use KD codes to reduce embedding size. Also, why the distilling in Chen et al., 2018b is a problem?
4. Did you run all experiments just one time? There is no confidence interval.
5. DPQ models have different compression ratios depending on the size of K and D. It would be great to show the change in PPL according to the compression ratio of DPQ models.
6. Can we apply it to pre-trained models like BERT?

**Experience Assessment:**

I have read many papers in this area.

**Review Assessment: Checking Correctness Of Derivations And Theory:**

I assessed the sensibility of the derivations and theory.

**Review Assessment: Checking Correctness Of Experiments:**

I assessed the sensibility of the experiments.

**Review Assessment: Thoroughness In Paper Reading:**

I made a quick assessment of this paper.

---

> ### Author Response · Authors · 2019-11-14
> **Response**
>
> Thank you for your time and the detailed comments! Please see below for clarifications and more empirical evidence.
>
> [Significance and Novelty]
>
> We believe in the significance of embedding compression for inference/deployment time, because embedding parameters make up a large part of model parameters in a wide variety of models. For example, 95% of parameters in the medium-sized LSTM (refer to Paragraph 1 of Section 1) are embedding parameters; 99% in the MLP-based model for text classification, 22% in the BERT-base model, etc.
>
> To our best knowledge, previous efforts offer up to ~50x compression ratios without performance loss, while our work achieve SOTA compression ratios of up to ~200x on 10 datasets across three different language tasks (we also have new results on BERT in Section H of the Appendix). We argue that these are empirically significant results.
>
> Novel contributions are made compared to Chen et al., 2018b's, and these contributions led to performance improvement by very large margins (Table 3 and 4). Their work and our work present very different methodologies. Here we list the major differences:
>
>     1) In their work, discrete codes are directly associated with each of the symbols; in this work, discrete codes are computed as outcome of product quantization.
>     2) Our formulation of discrete codes with product quantization allows us to derive two variants with different approximation techniques (softmax-based and vector quantization-based).
>     3) Our method uses a novel composition function (inspired by product quantization), which is much more efficient than before (smaller memory footprint and less computation time overhead, Figure 4).
>
> [Experiments and Why Discrete Codes (aka KD codes)]
>
> Re “there's been a lack of explanation as to why this should be done only through KD codes”.
>
> First of all, we have added more comparisons to conventional methods in the Appendix G. Table 9 shows comparisons with scalar quantization, product quantization, low-rank factorization, as well as other discrete code baselines (Shu and Nakayama 2017 & Chen 2018b) for language modeling. Table 10 shows similar comparisons on text classification tasks. These results show that previous methods struggle to maintain task performance when trying to achieve good compression ratios, and our method DPQ outperforms them by large margins.
>
> Then, we provide more analysis on why our methods work better. Unlike traditional quantization techniques that accumulate and amplify quantization error, our method makes it end-to-end differentiable so that the neural nets can adapt to quantization error. DPQ also relates to factorization-based method, but DPQ can produce high-rank embedding table with sparse factorization, so it is more expressive (Theorem 1 in Section 2.1). More analysis with conventional approaches is in Appendix F.
>
> Re “It would be great to show the change in PPL according to the compression ratio of DPQ models.”
>
> Figure 3, 7 and 8 show the trade-offs between task performance (PPL, BLEU or accuracy) and compression ratios for different sizes of K and D.
>
> Re “Can we apply it to pre-trained models like BERT?”
>
> Yes. We added experiments on BERT (Appendix H). Without any hyperparameter tuning, DPQ could compress the embedding layer 37x with negligible loss in performance.
>
> Re “Did you run all experiments just one time? There is no confidence interval.”
>
> Some of our experiments are computationally expensive (e.g. days-long). However we did repeat experiments where resources allowed, e.g. the PTB language modeling experiments and the WMT’19 En-De experiments, and found that the results were stable (e.g. std=0.6 over 4 runs for PPL in PTB LM). In the paper, we follow recent evaluation protocol in these tasks (e.g. [Shu and Nakayama 2017, Vaswani et al 2017]) and left out confidence intervals.
>
> [Other comments]
>
> Re “why the distilling in Chen et al., 2018b is a problem?”
>
> Distillation leads to more computation cost and in practice a more complex pipeline. Training with distillation requires pre-training of the embedding layer, which means the same model has to be trained twice (more computation). We also have to maintain two embedding tables for distillation (more memory).
>
> Re “It is usually not necessary to train the entire embedding vector on GPU, so it would not be a big issue in the actual learning process.”
>
> Our goal is to reduce the embedding table size at the inference/deployment stage. E.g. we would be able to compress big models so that they can be deployed on mobile devices. It is not the goal of this work to improve training of embeddings.
>
> We hope we have provided better explanation and more evidence for the contribution of this work, and are happy to address any further concerns!

---

### Author Response · Authors · 2019-11-14
**Revision**

We have updated the paper with the following changes:

    1) We provided detailed analytical comparisons between our work and Chen et al 2018b and other traditional compression techniques in Appendix F as suggested by Reviewer #1 and #2.
    2) We added more empirical comparisons of our work with a broader set of baselines on more tasks in Appendix G as suggested by Reviewer #2 and #3.
    3) We revised Theorem 1 and made some clarifications as suggested by Reviewer #3.
    4) We added new results on BERT compression in Appendix H.

---

### Decision · Program_Chairs · 2019-12-19

**Decision:**

Reject

**Comment:**

The presented paper gives a differentiable product quantization framework to compress embedding and support the claim by experiments (the supporting materials are as large as the paper itself). Reviewers agreed that the idea is simple is interesting, and also nice and positive discussion appeared. However, the main limiting factor is the small novelty over Chen 2018b, and I agree with that. Also, the comparison with low rank is rather formal: of course it would be of full rank , as the authors claim in the answer, but looking at singular values is needed to make this claim. Also, one can use low-rank tensor factorization to compress embeddings, and this can be compared.
To summarize, I think the contribution is not enough to be accepted.